# The Impact of COVID-19 on the Preparation for the Tokyo Olympics: A Comprehensive Performance Assessment of Top Swimmers

**DOI:** 10.3390/ijerph18189770

**Published:** 2021-09-16

**Authors:** Emese Csulak, Árpád Petrov, Tímea Kováts, Márton Tokodi, Bálint Lakatos, Attila Kovács, Levente Staub, Ferenc Imre Suhai, Erzsébet Liliána Szabó, Zsófia Dohy, Hajnalka Vágó, Dávid Becker, Veronika Müller, Nóra Sydó, Béla Merkely

**Affiliations:** 1Heart and Vascular Center, Semmelweis University, 1122 Budapest, Hungary; csulak.emese@gmail.com (E.C.); timea.kovats@gmail.com (T.K.); tokmarton@gmail.com (M.T.); lakatosbalintka@gmail.com (B.L.); kovatti@gmail.com (A.K.); suhaiimi987@gmail.com (F.I.S.); sz.liliana.e@gmail.com (E.L.S.); dohyzsofi@gmail.com (Z.D.); vagoha@gmail.com (H.V.); becdavid@gmail.com (D.B.); merkely.bela@gmail.com (B.M.); 2Hungarian Coaches Association, 1146 Budapest, Hungary; petrov.arpad@gmail.com; 3Department of Sports Medicine, Semmelweis University, 1122 Budapest, Hungary; 4Argus Cognitive, Inc., Lebanon, NH 03766-1441, USA; staub.levente@gmail.com; 5Pulmonology Clinic, Semmelweis University, 1083 Budapest, Hungary; muller.veronika@med.semmelweis-univ.hu

**Keywords:** athlete, swimming, COVID-19, cardiopulmonary exercise test, performance assessment

## Abstract

Background: The Olympic preparation of athletes has been highly influenced by COVID and post-COVID syndrome. As the complex screening of athletes is essential for safe and successful sports, we aimed to repeat the 2019-year sports cardiology screening of the Olympic Swim Team before the Olympics and to compare the results of COVID and non-COVID athletes. Methods: Patient history, electrocardiogram, laboratory tests, body composition analysis, echocardiography, cardiopulmonary exercise test (CPET) were performed. We used time-ranking points to compare swimming performance. Results: From April 2019, we examined 46 elite swimmers (24 ± 4 years). Fourteen swimmers had COVID infection; all cases were mild. During CPET there was no difference in the performance of COVID (male: VO_2_ max 55 ± 4 vs. 56.5 ± 5 mL/kg/min, *p* = 0.53; female: VO_2_ max 54.6 ± 4 vs. 56 ± 5.5 mL/kg/min, *p* = 0.86) vs. non-COVID athletes (male VO_2_ max 56.7 ± 5 vs. 55.5 ± 4.5 mL/kg/min, *p* = 0.50; female 49.6 ± 3 vs. 50.7 ± 2.6 mL/kg/min, *p* = 0.47) between 2019 and 2021. When comparing the time results of the National Championships, 54.8% of the athletes showed an improvement (*p* = 0.75). Conclusions: COVID infection with short-term detraining did not affect the performance of well-trained swimmers. According to our results, the COVID pandemic did not impair the effectiveness of the preparation for the Tokyo Olympics.

## 1. Introduction

Comprehensive sports cardiology screening is essential to detect potentially harmful cardiovascular disorders, but with repeated exams, it is also useful to assess and follow-up athletic performance [1]. Besides the pre-participation screening exams (history, physical exam, ECG), a detailed athletic screening also includes laboratory exam, echocardiography, and cardiopulmonary exercise test (CPET), which is the best way to measure cardiorespiratory fitness and maximal exercise tolerance in a clinical environment [2].

The breakout of COVID-19 has changed not only the life of athletes but also the whole healthcare service. The pandemic of COVID-19 brought uncertainty in athletes’ life regarding the ability to practice, the safety of training, quarantine regulations, canceling or postponing athletic events, including the 2020 Olympics [3,4,5]. Due to the interrupted preparation, sport adaptation could be impacted with fears of a deterioration in performance. Moreover, during the COVID-19 pandemic, only emergency care was available, performance testing such as CPET was postponed [6,7,8].

Swimming is potentially more affected than other sports, as training opportunities have become more complicated with the temporary closure of swimming pools [9]. With this in mind, at the beginning of the year 2020, when there was no final decision about postponing the games to 2021, some predictions also emerged the current year will not register world records in swimming for the first since 1986 [10]. Although swimming is an individual sport, the possibility of spreading the virus is high because elite athletes usually train and frequently live together.

Athletes infected with COVID-19 have to face various acute symptoms and may experience long COVID-19 syndrome. Even though COVID-19 mainly manifests in the respiratory system and causes mild to severe viral pneumonia, multiple cardiovascular sequelae can occur, like myocarditis, increased coagulopathy, pulmonary embolism, or even cardiogenic shock [11,12]. In athletes, COVID-19 infection may occur without any symptoms or mild disease with loss of smell and taste, though subclinical myocardial injury could be observed, which can lead to serious complications [12]. To exclude the subclinical myocardial injury a detailed cardiology examination is required even in asymptomatic COVID-19 positive athletes before they return-to-play [13,14,15]. Usually, the post-COVID examination protocol does not contain an exercise test only after negative results of baseline exams (resting electrocardiogram (ECG), laboratory test, and echocardiography) [16,17,18]. However, a cardiopulmonary exercise test is often performed to assess performance and can thus be applied to post-COVID athletes to detect effects on exercise capacity [17].

The present prospective observational study aimed to present and discuss the results of two consecutive comprehensive sports cardiology evaluations of national swimmers preparing for the Tokyo Olympics. We focused on the cardiac adaptation assessed by echocardiography and sports performance using CPET and swimming performance determined by time-ranking points. After the COVID-19 break-out, we also aimed to determine the impact of COVID-19 infection on the preparation of athletes and to compare the performance of COVID and non-COVID athletes.

## 2. Materials and Methods

### 2.1. Study Design

In our prospective observational study, we performed detailed sports cardiology screening on elite swimmers during identical times in the 2019 and 2021 training programs. Athletes were grouped and analyzed according to the COVID-19 status: COVID and non-COVID athletes. Ethical approval was obtained from the Central Ethics Committee of Hungary (athletic screening-ETT TUKEB IV/10282-1/2020/EKU and post-COVID screening-ETT TUKEB IV/9697-1/2020/EKU) and, all participants provided their informed consent.

### 2.2. Participants

All the swimmers were members of the National Swim Team Hungary with qualification for the Tokyo 2020 Olympic Games. A general and common practice in high-performance swimming is the so-called periodization. The four-year Olympic training plan is structured by macrocycles (seasonal or one-year-long training programs) and mesocycles (shorter, 6–8 weeks training periods) containing different training units. The aim of this complex and long task plan is to bring the athletes to their peak performance at the main national and international events, with their highest peak at the Olympics [19].

In Hungary, the 2019 and the 2021 calendar was similar due to the COVID-19 pandemic. The winter seasons ended with the major national event of the Hungarian Championship in March, and the summer seasons ended with the World Championship in 2019 and the Olympic Games in 2021. The goal of the Hungarian Championships was in both seasons the qualification for the annual world-class events, therefore we can assume that all the swimmers were in a similar phase of preparation in both screening periods [20].

Before the sports cardiology screening, athletes were instructed to avoid both swimming and dryland training 24 h before the evaluations. Nevertheless, they were instructed to do their morning routine including having breakfast and staying hydrated.

### 2.3. Procedures

#### 2.3.1. Sports Cardiology Screening Protocol

We assessed the swimmer’s personal history and symptoms on a sport-specific questionnaire. Standard resting 12-lead ECG were recorded (CardioSoft PC, General Electric Healthcare, Helsinki, Finland). Detailed ECG analysis was performed using the 2018 recommendations for ECG interpretation of athletes by the European Society of Cardiology [21].

We performed a detailed laboratory test for all the athletes. Besides the routine laboratory parameters (qualitative and quantitative blood count, renal and liver function, C-reactive protein, ions, creatine kinase), iron panel (ferritin, total iron-binding capacity, transferrin, transferrin saturation) and vitamin D levels were also checked. Iron deficiency was defined if the ferritin level was lower than 100 μg/L, without any laboratory or clinical sign of infection. Vitamin D was considered low below <50 ng/mL. In 2021, due to the COVID-19 safety regulations, COVID-19 antigen tests (Panbio™ COVID-19 Ag Rapid Test Device, Abbott; Genedia W COVID-19 Rapid Ag test, GCMS, Jena, Germany,) were performed before the examination. We also complemented the laboratory tests with spike-protein COVID-19 antibody tests.

Before the body composition analysis, we measured the athlete’s weight and height with an automatic body mass index measuring stadiometer InBody BSM370 (InBody Co. Ltd., Seoul, Korea). Body composition analyses (body mass, percentage of fat, percentage of muscle) were carried out with InBody 770 based on bioelectric impedance analysis (InBody 770, InBody Co. Ltd., Seoul, Korea).

Echocardiography examinations were performed with a commercially available ultrasound system (General Electric Vingmed Ultrasound E95 Ultrasound system, 4Vc-D probe, Horten, Norway). According to current guidelines, a standard acquisition protocol consisting of 2D loops from parasternal, apical, and subxiphoid views was applied [22]. Left ventricular septal and posterior wall thicknesses, end-diastolic diameter, ejection fraction, and right ventricular basal end-diastolic diameter were also quantified.

Cardiorespiratory fitness was determined by a vita maxima incremental test on a treadmill ergometer (General Electric T-2100, Healthcare, Helsinki, Finland). (Preparation phase: 1 min standing position, 2 min warm-up walk at 6 km/h followed by running at 8 km/h with a progressive workload increment rate of 1.5% every 120 s until exhaustion and a period of 1-min active recovery and 4 min passive recovery). Athletes were instructed not to hold the handrail. Heart rate recovery was calculated from peak heart rate minus 1 min recovery heart rate [23]. Gas volume and composition analysis were performed using a breath-by-breath automated cardiopulmonary exercise system (Respiratory Ergostik, Geratherm, Bad Kissingen, Germany). The subjects were encouraged to achieve maximal effort, which was confirmed by respiratory exchange ratio (RER), and also by reaching the predicted maximal heart rate and a plateau in maximal oxygen uptake. Evaluated variables: treadmill time (min), maximal oxygen uptake (VO_2_ max; L/min and mL/kg/min), respiratory exchange ratio, ventilation (VE; L/min), ventilatory equivalent for carbon dioxide production (VE/VCO_2_), oxygen uptake milliliters per heartbeat-O_2_ pulse (L/min), lactate (mmol/L) measurements every 2 min during the test from the fingertip and after 5 min in the recovery phase.

#### 2.3.2. Post-COVID Return-to-Play Examinations

The whole team were instructed to indicate if somebody has any symptoms. After a COVID-related symptom appeared, a PCR test was immediately performed at our Clinic. Therefore, the detection and evaluation time was usually less than 24 h. After COVID-19 infection, a “return-to-play” examination was required to resume training. All the athletes were examined after their 10–14 days of the quarantine period. A COVID-specific questionnaire was filled out about the length and the symptoms of the disease. Detailed patient history and training habits were also obtained. Laboratory tests were performed containing high sensitive Troponin T and d-dimer. Physical examination, 12-lead resting ECG, echocardiography, and body composition analysis were carried out. A 24-h Holter ECG was applied in case of palpitation. In case of abnormalities in the baseline tests or prolonged complaints chest computer tomography (CT) and cardiac magnetic resonance imaging (MRI) were performed.

#### 2.3.3. Time-Ranking Points Calculation

To assess personal performance, we used the time-ranking points given by the International Swimming Federation (FINA). These points allow us to compare the time results of different events. The time-ranking points table assigns point values to swimming performances. More points signal faster performances (world-class performances are typically 950 points or higher), and fewer points show slower performances. Point values are recalculated every year, and the table is named according to the year when the base times were defined. In our study, we used the “FINA Point Scoring 2019” table. The base times for this particular table are the latest long course world record in each event (female and male), approved by FINA until the end of the respective year (31 December). Time-ranking points are calculated using a cubic curve. With the swim time (T) and the base time (B) in seconds, the points (P) are calculated with the following formula: P = 1000 × (B/T)^3^. The exact formula is used to calculate points from times. Then all point values are truncated to the integer number [24].

### 2.4. Statistical Analysis

Statistical analyses were performed using the GraphPad Prism 8.0.1 program package. The Shapiro-Wilk’s test was used to test the normality of variables. Normal distribution data are presented as mean ± standard deviation, while non-normal distribution data are presented as median with interquartile range. Categorical variables are presented as percentages and frequencies. Comparisons of the means of continuous variables were performed using unpaired t-tests. The distributions of non-normal continuous variables were compared by Mann-Whitney tests. *p*-value < 0.05 was considered to indicate statistical significance.

## 3. Results

### 3.1. Study Population

Comprehensive sports cardiology screening was applied to 46 swimmers. All the athletes were members of the National Swim Team Hungary and were qualified for the Tokyo 2020 Olympic Games. They have been training for more than 20 h/week for at least 13 years in different swimming clubs throughout the country. No positive family history or risk factor was revealed for cardiovascular diseases.

Out of the 46 athletes 14 had COVID-19 infection between March 2020 and April 2021. Out of them, ten swimmers had PCR-confirmed COVID-19 infection, while four swimmers had positive antibody tests. There was no difference in age, sports experience, and pre- or post-COVID training hours between non-COVID and COVID swimmers (Table 1). None of the athletes was hospitalized due to a COVID-19 infection; only one athlete needed medication. However, 10–14 days quarantine was required in every case, thus the athletes could not train for an average of 24 days.

### 3.2. Findings of the Sports Cardiology Screening

During the comprehensive sports cardiology screening, most of the alterations were recognized during the 2019 screening period. No pathological ECG or echocardiography finding was confirmed during the exams. In 2019 screening three 24-h ambulatory blood pressure monitoring exams (ABPM) were performed, in one case angiotensin receptor blocker was initiated. The 2021 control showed normal blood pressure values in all the athletes.

Regarding the laboratory exams, no major alteration was found regarding the hemoglobin, hematocrit, renal, and liver function. There was no difference between COVID and non-COVID athletes. Creatine-kinase (CK) increase was frequent as a result of training. There was no difference in CK between the non-COVID and COVID groups (female non-COVID vs. female COVID: 190 ± 126 U/L vs. 134 ± 76 U/L, *p* = 0.14; male non-COVID 230 ± 155 U/L vs. male COVID 230 ± 205 U/L, *p* = 1.0).

Iron panel and vitamin D levels were checked in all the athletes in both screening sessions. The frequency of iron deficiency was 53% in non-COVID athletes and 50% in COVID athletes (*p* = 0.2). The average ferritin level in non-COVID female athletes was 63.4 ± 42.3 μg/L and in COVID female athletes were 60.4 ± 32.3 μg/L (*p* = 0.84). In non-COVID female (68 ± 45.6 μg/L vs. 58 ± 39.5 μg/L, *p* = 0.6) and male (124.4 ± 62.2 μg/L vs. 113.8 ± 41 μg/L) athletes from 2019 to 2021, the ferritin level showed a decreasing tendency without significant difference between the two years. Low vitamin D level was observed in 64% of COVID athletes and 50% of non-COVID athletes. Vitamin D level was lower in COVID infected athletes (47.7 ± 11.0 ng/mL vs. 37.8 ± 11.0 ng/mL, *p* < 0.05).

Regarding body composition exams, no difference was verified between female and male COVID and non-COVID athletes in muscle mass and fat mass percent. From 2019 to 2021, both COVID and non-COVID athletes maintained their muscle mass and showed improving tendency regarding body fat percent.

Echocardiography results are shown in Table 2. Ejection fraction showed an improving tendency from 2019 to 2021 both in COVID and non-COVID athletes. Left ventricular posterior wall thickness improved in all groups, while no significant difference was detected between the COVID and non-COVID groups. Left ventricular end-diastolic diameters showed a decreasing tendency, with increased right ventricular diameter in all groups.

In the CPET exams, all athletes gave good effort by reaching at least 1.10 in the respiratory exchange ratio. No significant arrhythmias or ST changes occurred during any exercise test.

CPET values were typical for the elite-level swimmers. Treadmill time and maximum load values were similar in the two groups; no relevant difference was verified. Maximal aerobic capacity changes were not significant, neither from 2019 to 2021 nor between COVID and non-COVID athletes. Improving tendency in VO_2_ max could be observed in female non-COVID and male COVID athletes. Ventilation/carbon dioxide production nadir and peak lactate did not differ in COVID, and non-COVID athletes, their changes between 2019 and 2021 were not relevant (Table 3).

### 3.3. Performance Analyses by Time-Ranking Points

Overall, the athletes’ performance improved by 54.8% from 2019 to 2021 with no regard to COVID-19 status. In COVID athletes, the improvement was 55.6%. Non-COVID athletes reached a 54.5% improvement in ranking points. There were no significant differences between COVID and non-COVID athlete’s time-ranking points (*p* = 0.75) (Figure 1).

### 3.4. Findings of Return-to-PLAY Examination

According to the COVID-19 specific questionnaire, 86% of COVID athletes had symptoms during the infection. The most common symptoms were fatigue (64%), fever (50%), muscle ache (43%), and dyspnoea (43%). Less common symptoms of COVID-19 were: headache (36%), nasal congestion, runny nose (36%), sleep disorders (36%), cough (21%), anosmia, ageusia (21%), throat ache (21%), and palpitation (7%).

Post-COVID laboratory test verified in female athletes: 172 ± 146 μg/L ferritin and 39 ± 15 ng/mL vitamin D level and in male: 105 ± 17 μg/L ferritin and 27 ± 7 ng/mL vitamin D level.

A 27-year-old female athlete had atypical chest pain with coughing, running nose, shortness of breath, and fever. She reported her symptoms immediately after she had experienced them. The first reverse transcriptase-polymerase chain reaction (rt-PCR) assay was negative and the symptoms remained; after a few days, we repeated rt-PCR, which was positive. She started the 14-day quarantine. Due to the complaints, we also performed a chest CT, which showed COVID-19 pneumonia (Figure 2). Antibiotic (azithromycin 500 mg for three days) and antipyretic (paracetamol 400 mg 2 × 1 for five days) therapy were administered. After four weeks of detraining a control CT was performed, pulmonary infiltration regressed, only discrete reticulation could be observed (Figure 3). She started her training after the control CT exam with a slow increase in training intensity and duration. She has not reported any further symptoms or exercise-related complaints.

In two antibody-revealed COVID-19 infections, cardiac MRI was performed due to elevated high sensitive Troponin T levels (16 ng/L, 30 ng/L), no abnormality was found.

In summary, no other pathological abnormalities were confirmed during the examinations. All athletes could re-start the training.

## 4. Discussion

Our study shows that mildly symptomatic COVID-19 infection did not affect cardiac adaptation or performance in elite swimmers assessed by echocardiography and CPET. Moreover, those athletes who could not train due to COVID-19 infection were able to maintain muscle and did not gain fat. Iron deficiency and low vitamin D levels were frequent in both COVID and non-COVID athletes. Vitamin D levels were lower in COVID-infected athletes.

We performed a comprehensive sports cardiology screening including ECG, laboratory test, body composition, echocardiography, and CPET on the members of the National Swim Team Hungary in 2019 and 2021. The two similar screening protocols allowed us to compare the sport adaptation and performance parameters of COVID and non-COVID athletes, and to assess the effect of COVID-19 on the preparation for the Olympic Games.

All fourteen COVID-19 cases (28%) were considered mild, cardiac MRI and chest CT were only required in three cases, all athletes could return-to-play after approximately three weeks. In a study by Italian researchers, ninety asymptomatic or mildly symptomatic competitive athletes were involved in different sports. Cardiac consequences of COVID-19 infection were investigated by detailed cardiovascular screening (physical examination, blood tests, spirometry, 12-lead resting ECG, 24-h ambulatory ECG monitoring, echocardiography, CPET) [25]. Perimyocarditis was found in one case and pericarditis in two cases, instances which are as low as in our study.

We found significantly lower vitamin D levels in COVID-infected athletes. This correlates with several recent study results, as higher vitamin D levels are associated with the reduced risk and severity of COVID-19 infection [26,27]. In a review that summarizes the results of more than thirty observational studies, it is stated that athletes had better athletic performance, better health, and reduced risk for COVID-19 by maintaining serum vitamin D concentrations above 40 ng/mL. Vitamin D cut-off values are varying according to the patient populations and regional differences. In our study, we considered 50 ng/mL or higher vitamin D levels as normal, in line with one of the first reviews of athletic vitamin D research [28].

In our study, all the athletes underwent echocardiography exams twice to assess cardiac adaptation. However, as all of our athletes’ baseline echocardiography showed marked morphological sport adaptation, further signs of wall thickening or chamber dilatation would not be expected. In a cross-sectional study, which compares echocardiographic data of aquatic sports athletes (including twenty males and seventeen female swimmers), chamber diameters and wall thicknesses were similar to our data [29]. During the two consecutive echocardiography exams, we found only a concentric hypertrophy tendency, while left ventricular end-diastolic internal diameters decreased and right ventricular diameters increased. As swimming is an endurance sport, both diameter and wall thickness increase would be expected, but nowadays, dryland strength training has become more widespread, thus static elements of the training are not negligible [30]. Our athletes also had more dryland strength training during the COVID-19 lockdown, which may have contributed to our results.

The other main focus of our study is the effect of COVID-19 on sports performance. Even short-term (<4 weeks) detraining can lead to the loss of exercise-induced adaptation mechanisms [31]. As a result of short-term detraining the maximal oxygen uptake and ventilatory efficiency decline, while respiratory exchange ratio and lactate levels increase [31,32,33]. According to our results, there was no difference in CPET parameters between non-COVID and COVID athletes, which means that no long-term effect of detraining was verified. A study that examined sixteen first division volleyball players concluded that athletes experienced typical consequences of three weeks of detraining, while their VO_2_ max values were normal. They were unable to estimate the effect of detraining on their performance, as they had had no baseline examination before COVID-19 infection. They also did not have a non-COVID control group at the same training status [34]. A study investigated the effect of an eight-week lockdown period on ten elite male handball players. They compared their performance to a baseline pre-season test performed in 2019. Athletes were instructed to train at home during the lockdown. They performed an incremental exertion test two times, in 2019 pre-season and 2020 after the eight-week of lockdown. They concluded that calculated power output and VO_2max_ did not differ between the pre-season and after the COVID-19 lockdown in 2020. However, shuttle run tests showed lower endurance capacity. Even though this study also has a baseline examination, they compare their results to a pre-season examination, not to a peak performance status, without CPET [35]. In another study by Dauty et al., the effect of COVID-19 quarantine on training has been assessed in twenty-nine high-level adolescent soccer players [36]. The athletes were not tested for COVID-19; therefore, they were not proved infected or patients with a medical condition. During the 2-month quarantine, they were instructed to perform physical exercises on peak heart rate lower than 80% predicted in their homes. They did not have CPET measurement, aerobic capacity was calculated according to a Yo-Yo field test. A decreased aerobic capacity was found, however, this may be the result of the long quarantine period and the submaximal training intensity [36]. Another study examined the effect of 40 days of COVID-19 quarantine period on the performance using a Yo-Yo test in twenty soccer players. They concluded that even though the athletes were training at home three times a week at 65–75% of maximal heart rate, the 40 days of quarantine had an impact on maximal speed, high-intensity running distance, acceleration, and deceleration. However, their maximal aerobic capacity calculated from the Yo-Yo test did not change [37]. To further assess swimming performance, we used time-ranking points to compare the swimming results of the national championships. There was no difference in ranking points between non-COVID and COVID athletes, which means that all the athletes improved their time results.

In our study, we found no difference in CPET parameters between the years 2019 and 2021. Our athletes were in their best personal condition in 2019, because their main competition was the 2019 Gwangju World Aquatics Championships. Then they had to achieve the same or better performance for the Tokyo Olympics, 2021.

### 4.1. Practical Implications

With our study, we try to emphasize the importance of regular comprehensive sports cardiology screening with echocardiography and CPET. Besides the screening of pathological disorders, sport adaptation and performance could be assessed and followed up with regular exams.

Factors affecting sports performance, such as iron deficiency and low vitamin D levels should be screened during the laboratory test and supplementation has to be initiated.

The consequences of COVID-19 should be screened during the return-to-play examination to allow safe sport activity. Mild COVID-19 illness generally does not trigger long-term performance decrement and most athletes recover well from COVID-19 disease.

### 4.2. Strengths and Limitations

Our study population is a homogeneous group that contains both female and male top-level swimmers with the same training intensity. All the athletes participated in the Tokyo Games. There is no other currently available detailed study examining the effect of COVID-19 on the performance of swimmers. This is the first study that compares the post-COVID performance of Olympic swimmers to a baseline pre-COVID examination and compares it with a non-COVID control group. We also used time-ranking points to compare not only endurance but also swimming performance. This allows us to compare the swimming time results in different strokes and distances.

The main limitation of the study is the low number of COVID-19 infected athletes in contrast to non-COVID athletes. The COVID-19 infected group only contains mild cases without hospitalization or a prolonged detraining period.

Another limitation of the study is that exercise tests were conducted in a clinical environment, which differs from swimmers’ own environment. Cardiopulmonary exercise tests were performed on a treadmill ergometer. For a swimmer, a cycle ergometer would be more appropriate.

In our study, we found statistically significant changes in echocardiographic parameters. However, these changes could be a result of inter-observer variability rather than real sport adaptation changes.

## 5. Conclusions

A comprehensive examination is useful in elite athletes not only to screen cardiac abnormalities after COVID-19 infection but also to assess sport adaptation and performance. Although cardiac consequences of COVID-19 infection are low, return-to-play post-COVID exam of athletes is essential for safe and successful sports. According to our results, we can conclude that short-term detraining and COVID-19 infection did not affect performance in well-trained swimmers in Hungary. According to the time results of the Hungarian National Championships, the COVID-19 pandemic did not impair the effectiveness of the preparation for the Tokyo Olympics.

## Figures and Tables

**Figure 1 ijerph-18-09770-f001:**
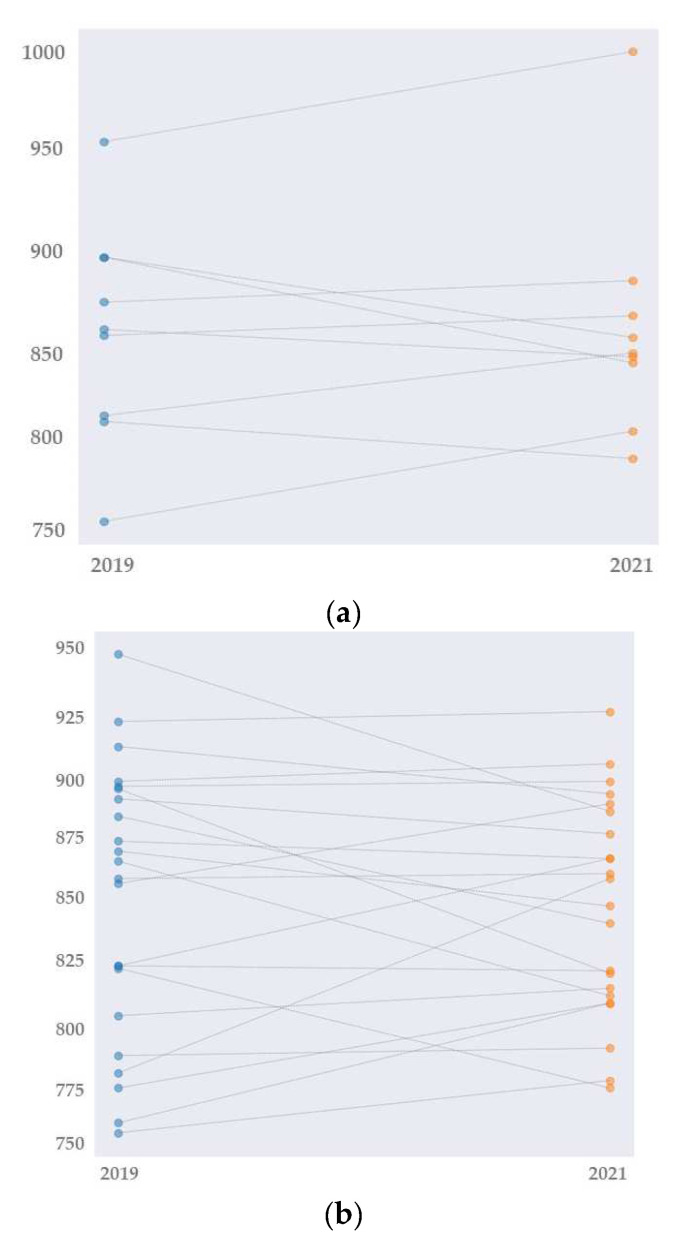
Time-ranking points change from 2019 to 2021 in COVID-athletes (**a**) and non-COVID athletes (**b**). Figure Legend: blue dots: time-ranking points of 2019, orange dots: time-ranking points of 2021.

**Figure 2 ijerph-18-09770-f002:**
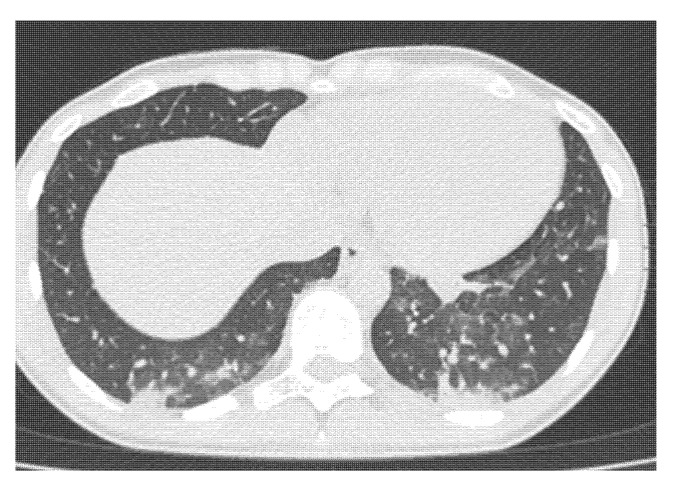
Baseline chest CT of a 27-year-old athlete. Confluent ground-glass opacities, pulmonary infiltrates.

**Figure 3 ijerph-18-09770-f003:**
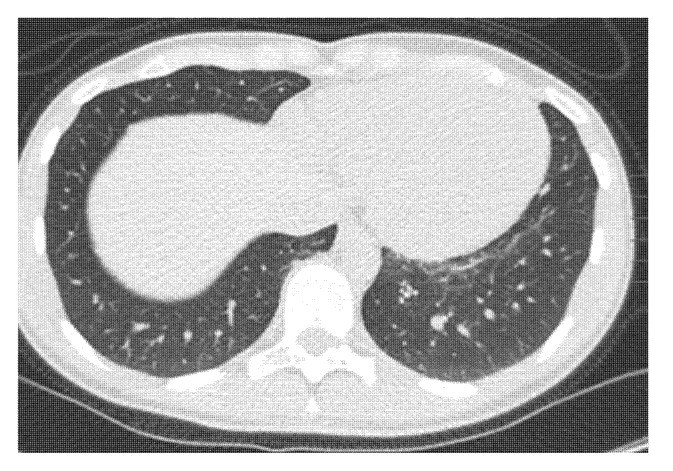
Control chest CT: regression, discrete reticulation.

**Table 1 ijerph-18-09770-t001:** Baseline characteristics according to COVID-19 infection.

	Non-COVID Swimmers (n = 32)	COVID Swimmers (n = 14)	P
Age (years)	24.2 ± 4.4	23 ± 3.8	0.40
Female (n, %)	14 (43.8%)	7 (50%)	0.70
Sports experience (years)	17.8 ± 4.5	16.8 ± 3.9	0.44
Pre-COVID training hours/week	24 ± 4.5	24.5 ± 3.9	0.71
Post-COVID training hours/week	24 ± 4.5	24.5 ± 3.9	0.71
Missed training days	18.5 ± 3.4	19 ± 6	0.76

**Table 2 ijerph-18-09770-t002:** Echocardiography results.

		Non-COVID Swimmers (n = 32)			COVID Swimmers (n = 14)			2019 vs. 2019	2021 vs. 2021
		2019	2021	*p*	2019	2021	*p*	*p*	*p*
Ejection fraction (%)	M	55.8 ± 2.1	58.8 ± 4.1	0.02 *	59.7 ± 8.0	59.3 ± 4.8	0.93	0.1	0.81
	F	59.1 ± 2.6	60.0 ± 3.3	0.48	56.9 ± 3.3	60.0 ± 4.7	0.17	0.097	1
Septal wall thickness (mm)	M	11 ± 1.2	10.8 ± 1.1	0.62	11.3 ± 2.5	11.4 ± 1.3	0.94	0.72	0.25
	F	9.2 ± 0.8	9.2 ± 1.3	0.97	9.9 ± 0.9	10.1 ± 0.7	0.52	0.11	0.15
Posterior wall thickness (mm)	M	9.6 ± 0.99	10.7 ± 1.1	0.009 *	9 ± 0	10 (10–11)	0.002 **	0.32	0.36
	F	8.4 ± 0.85	9.9 ± 1.3	0.003 *	8.4 ± 1	11 ± 1.4	0.002 **	1	0.12
LV end-diastolic diameter (mm)	M	55.7 ± 4.1	51.4 ± 2.9	0.003 *	54.7 ± 4.7	50.1 ± 3.5	0.14	0.71	0.02 *
	F	48.4 ± 2.3	44.4 ± 2.8	0.001 *	49 ± 2.5	46.4 ± 2	0.046 *	0.48	0.12
RV diameter (mm)	M	35.5 ± 3.3	37.9 ± 3.2	0.06	35.3 ± 4.2	39 (38–41)	0.24	0.93	0.20
	F	31.8 ± 4	34.6 ± 4.4	0.11	32.9 ± 3	37.6 ± 2.3	0.006 *	0.54	0.12

Legend-Abbreviation: M—male, F—female, LV—left ventricular, RV—right ventricular. Significance level: * *p* < 0.05, ** *p* < 0.005.

**Table 3 ijerph-18-09770-t003:** Cardiopulmonary exercise testing results.

		Non-COVID Swimmers (n = 32)			Covid Swimmers (n = 14)			2019 vs. 2019	2021 vs. 2021
		2019	2021	*p*	2019	2021	*p*	*p*	*p*
Resting HR (bpm)		68.4 ± 13.4	62.0 ± 11	0.06	69.0 ± 15	72.4 ± 17	0.61	0.90	0.024 *
Peak HR (bpm)		190.0 ± 10.3	190.5 ± 11.5	0.88	191.2 ± 9.1	188.0 ± 11	0.44	0.74	0.53
HR recovery (1/min)		32.3 ± 10.8	22.0 (20.0–32.0)	0.03 *	33.1 ± 13.9	23.5 (19.7–31.2)	0.32	0.85	0.78
RER		1.15 ± 0.05	1.17 ± 0.08	0.28	1.15 ± 0.07	1.17 ± 0.07	0.38	0.81	0.94
Treadmill time (min)	M	16.0 (13.0–16.5)	14.5 ± 2.7	0.12	14.5 ± 1.3	15.0 (13.7–15.0)	1.0	0.73	0.40
	F	13.2 ± 2.8	12.9 ± 1.6	0.80	14.4 ± 2.0	13.5 ± 2.0	0.45	0.35	0.55
Max load (Watt)	M	432.6 ± 74.0	402.9 ± 60.8	0.25	464.3 ± 25.8	458.0 ± 31.0	0.75	0.42	0.052
	F	294.0 ± 50.4	270.4 ± 41.3	0.32	311.4 ± 61.7	283.3 ± 41.5	0.37	0.52	0.59
VO_2_ max (L/min)	M	4.7 (4.5–5)	4.6 ± 0.7	0.89	4.7 ± 0.4	5.2 ± 0.6	0.14	0.89	0.08
	F	2.9 ± 0.3	3.1 ± 0.4	0.48	3.3 ± 0.5	3.2 ± 0.4	0.82	0.10	0.47
VO_2_ max (mL/min/kg)	M	56.7 ± 4.7	55.5 ± 4.5	0.49	55 ± 3.8	56.5 ± 4.9	0.53	0.41	0.20
	F	49.6 ± 3	50.7 ± 2.6	0.47	53.1 ± 5.5	52.9 ± 4.1	0.97	0.12	0.76
O_2_ pulse (mL/bpm)	M	25.0 ± 1.9	24.0 ± 1.8	0.23	25.6 ± 2.1	26.4 ± 2.4	0.58	0.58	0.03 *
	F	15.0 (13.7–17.9)	16.2 ± 2.1	0.64	17.8 ± 2.8	18.2 ± 2.7	0.82	0.06	0.18
VE (L/min)	M	158.0 ± 33.0	159.0 ± 31.5	0.95	153.0 ± 9.5	178.0 ± 16.6	0.03 *	0.74	0.19
	F	114.0 ± 17.0	109.0 ± 25.0	0.64	118.0 ± 10.4	111.0 ± 12.4	0.28	0.60	0.90
VE/VCO_2_	M	25.0 ± 2.2	29.4 ± 3.9	0.70	28.5 ± 2.2	28.9 ± 3.5	0.86	0.88	0.78
	F	32.9 ± 2	31.0 ± 2.9	0.12	31.5 ± 2.2	30.2 ± 1.6	0.32	0.17	0.58
Peak lactate (mmol/L)	M	7.6 ± 2.5	7.8 ± 3.1	0.88	9.7 ± 4.1	9.5 ± 2.1	0.95	0.22	0.23
	F	8.5 ± 2.5	9.1 ± 2.2	0.58	8.4 ± 1.9	6.6 ± 1.7	0.09	0.99	0.04

Legend-Abbreviation: M—male, F—female, HR—heart rate, BPM—beat per minute, RER—respiratory exchange ratio, VE—ventilation, MIN—minute. Significance level: * *p* < 0.05.

## Data Availability

The dataset presented in this study are available on request from the corresponding author. Due to patient’s data, privacy data are not made available publicly.

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
