# Peer review of "The Impact of COVID-19 on the Preparation for the Tokyo Olympics: A Comprehensive Performance Assessment of Top Swimmers"

_ijerph, 2021, doi:10.3390/ijerph18189770_

Round 1

Reviewer 1 Report

Dear authors, first of all, I would like to thank you for the effort and time dedicated to preparing and sending this manuscript, however, and after the decision to reconsider the acceptance of the manuscript after an important revision, it is appropriate to indicate some suggestions for improvement in the different sections of the article. Format and Rules of the journal: • Authors must follow the rules of the journal in relation to the format of tables • Authors must follow the journal's rules regarding the format of bibliographic references (Check all by modifying punctuation marks between authors and indicate the year of publication in bold) • Authors must specify in the manuscript the absence or absence of conflict of interest. • Authors must specify in the manuscript, the contribution that each of them has made (For the previous modifications: see web page / instructions for authors) Use of acronyms: Authors should review all acronyms in the manuscript because there are several, which the first time they appear in the text, they do not clarify their meaning. For example BMI or CPT Title: The title of the manuscript should reflect the content of the research more specifically. It should be more in line with the different objectives set.
Summary: This should not include abbreviations such as FINA unknown to most readers Introduction: Authors should improve the foundation, it is suggested that they do a literature review of the studies published in the last 5-10 years, thus updating the references of their text and that they incorporate a greater number of citations to give more solidity to their arguments. . Normally a foundation should have 6 to 8 paragraphs that place the reader in the subject under study and in this case it does not even reach the middle. Methodology and instruments: Authors must include in the manuscript: • The type and design of the study carried out • The study variables Results: the authors should improve the way of presenting the results. When describing the sample, it is necessary to indicate the number of women who participated in the study The results relative to table 2 are replicated both in table and written format. Authors should select from the table only those most outstanding results, thus, they will facilitate the reading of the manuscript. In addition, the tables must be adapted to the format requested by the journal. In their legend, the significance value chosen by the authors must be indicated. Discussion: The discussion should be a space where the authors discuss the results found in their work with other equal or similar studies. Although the study is novel, the authors must include studies published between the last 5-10 years in order to give more solidity to their arguments. Bibliographic references: put them in the format indicated by the journal.  

Reviewer 2 Report

This study aimed at evaluating national swimmers and the potential effect of COVID-19 infection. The study is focused on a relevant research area considering the global impact on pandemic, as COVID-19. A revision of the manuscript is required to increase the quality of the scientific and academic writing style.        

INTRODUCTION

Line 45-47. Quite unclear sentence. Please re-write it.

Line 47-48. Suggestion “Although swimming is an individual sport, the possibility…”

Line 50-52. Suggestion “Despite COVID-19 mainly manifests in the respiratory system and causes mild to severe viral pneumonia, multiple cardiovascular sequelae can occur, like myocarditis, increased coagulopathy, pulmonary embolism or even cardiogenic shock.”

Introduction is quite brief to clearly understand the gap in literature, the new research question, the novelty and originality of the current study. Moreover, hypotheses are not clearly stated. I suggest implementing the last part of introduction. Use past tense for the purpose of the study.

METHODS

In its current form, materials and methods section is not clear and sufficient. It should be revised following a clear structure of study design, participants, procedures, statistical analysis.

The study design has not been explained. Provide a section to describe the participants and their characteristics. Include point 2.1 2.2 and 2.3 in procedures.

Line 148-150. Move this sentence at the beginning of section in the new paragraph of study design.

Line 195. “ABPM” spell out first time.

RESULTS

Line 216, 218, 221, 226. What do the numbers between bracket represent? Please clarify.

DISCUSSION

Line 335-336. This statement requires a reference. Please add.

Line 344. Please add the reference here.

A paragraph for detailed practical implications should be provided.

Tables and reference list are not in accordance with the journal guideline. Please revise accordingly.

Author Response

Dear Reviewer 2,

Thank you!

Best regards,

Authors

Reviewer 3 Report

Dear Authors,

I want to congratulate a big effort to prepare a manuscript relevant for practitioners of sports medicine and coaches in pandemic times.

It is an important topic in the time of Olympic preparations in the Covid-19 pandemic.

Your paper is interesting and good written; nevertheless, I have a few comments, questions and suggestions.

Lines 59-62:

First of all, what was the aim of the study? - perform the test or determine the effect

Detailed suggestions:

Introduction:

Lines 47-48: However, swimming is an individual sport, the possibility of spreading the virus is high because elite athletes usually train together.

Why are those lines important, and why did you put this sentence here? Would you please add any literature or supportive explanation?

Line 55: detailed cardiology examination is required - in my opinion, additional information on what detailed examination is, is needed

Lines 56-58: Usually, the post-COVID examination protocol does not contain an 56 exercise test only after negative results of baseline exams [10-12]. However, to compare the 57 performance cardiopulmonary exercise test (CPET) is essential. - you haven't described usually protocols

Material and Methods:

Well described section, but in Section 2.2. in my opinion, the information how long after quarantine (positive PCR test) you have performed an examination, and in which part of the yearly training period. Suppose 2019 and 2021 testing were at a similar time in the season.

Results:

Lines 166-167: in the bracket description of the group is not complete (female)

Line 172-174 and Table 1: Concerne is Why did you put 14 COVID swimmers, not 10, with a positive PCR test as you described earlier?

Line 187-191: Is this information relevant to your study. Does the vaccination had an impact on experiment protocol

Line 211-212: It is an important real-life information

Line 229: ....however, in male COVID athletes, it did change - what was your idea here?

Line 234-235: right ventricular diameters are higher in women compared to men in endurance sport. Look at the

M. Konopka; K. Burkhard-Jagodzinska; W. Krol et al. (2015). Moderated Posters session: pulmonary hypertension and other conditions. Eur Heart J Cardiovasc Imaging Abstracts Supplement16(suppl_2), S67-S69.

Line 237 Table 2: I have doubts about ALL columns in the table. It can affect statistical results. Please reconsider if there is a necessity to present the mean value from the two measurements. In my opinion more important is the difference betweenmeasurements.

Line 252 Table 3: I am not sure if combining the man and woman results in treadmill time or max load are relevant. In my opinion, it should be presented separately. Once again, please explain the presence of the ALL column.

Lines 256-261 Paragraph 3.3: table or figure will be helpful to present FINA results and their changes

 Discussion:

Line 327-328: please explain more deeply how dryland change the hearth diameters (some examples and literature)

Line 338-339: According to our results, there was no difference in CPET parameters between non-COVID and COVID athletes, which means that no long-term effect of detraining was verified. - Please add additional information about the training period in the yearly plan. As we know performance is changing in the year, so it will be valuable information.

Lines 361-363 - look at the comment above.

Conclusion:

Lines 399-400: According to our results, we can conclude that short-term detraining and COVID-19 infection had no effect on performance in well trained swimmers in central Europe - in my opinion, this conclusion is not supported by the results and it is not an outcome of the present study. Reconsider rewriting it.

Author Response

Dear Reviewer 3,

Thank you!

Best regards,

Authors

Round 2

Reviewer 1 Report

Dear authors:
I thank you for the modifications made in the manuscript, since they provide solidity to the research for the consideration of publication in the journal.
Cheers.

Author Response

Dear Reviewer,

Thank you for your excellent work. Your review let us improve our mauscript.

Best regards,

Emese Csulak

Reviewer 3 Report

You did a great job, and your paper clarity improved.

Nevertheless, I have a few suggestions:

  1. Add additional explanation about time from detection/sickness to evaluation and if this period impacted the results in 4 additional "non-positive" athletes. If the time doesn't impact the outcome of the evaluation, it is worth writing it.
  2. I suggest adding Figure 1. - in my opinion, it will give an overview of the changes. Secondly, I think that adding information on the impact of Covid on performance in an athlete who undergoes medication can be important for the reader 

Author Response

Dear Reviewer,

Best regards,

Authors
